

# Pharmacophagy in green lacewings (Neuroptera: Chrysopidae: *Chrysopa* spp.)?

Jeffrey R. Aldrich[1], Kamal Chauhan[2] and Qing-He Zhang[3]

[1] Associate, Department of Entomology & Nematology, University of California, Davis, CA, United States
[2] Invasive Insect Biocontrol and Behavior Laboratory, United States Department of Agriculture-Agriculture Research Service, Beltsville, MD, United States
[3] Director of Research, Sterling International, Inc, Spokane, Washington, United States

Corresponding author
Jeffrey R. Aldrich,
drjeffaldrich@gmail.com

## ABSTRACT

Green lacewings (Neuroptera: Chrysopidae) are voracious predators of aphids and other small, soft-bodied insects and mites. Earlier, we identified (1*R*,2*S*,5*R*,8*R*)-iridodial from wild males of the goldeneyed lacewing, *Chrysopa oculata* Say, which is released from thousands of microscopic dermal glands on the abdominal sterna. Iridodial-baited traps attract *C. oculata* and other *Chrysopa* spp. males into traps, while females come to the vicinity of, but do not usually enter traps. Despite their healthy appearance and normal fertility, laboratory-reared *C. oculata* males do not produce iridodial. Surprisingly, goldeneyed lacewing males caught alive in iridodial-baited traps attempt to eat the lure and, in Asia, males of other *Chrysopa* species reportedly eat the native plant, *Actinidia polygama* (Siebold & Zucc.) Maxim. (Actinidiaceae) to obtain the monoterpenoid, neomatatabiol. These observations suggest that *Chrysopa* males must sequester exogenous natural iridoids in order to produce iridodial; we investigated this phenomenon in laboratory feeding studies. Lacewing adult males fed various monoterpenes reduced carbonyls to alcohols and saturated double bonds, but did not convert these compounds to iridodial. Only males fed the common aphid sex pheromone component, (1*R*,4a*S*,7*S*,7a*R*)-nepetalactol, produced (1*R*,2*S*,5*R*,8*R*)-iridodial. Furthermore, although *C. oculata* males fed the second common aphid sex pheromone component, (4a*S*,7*S*,7a*R*)-nepetalactone, did not produce iridodial, they did convert ∼75% of this compound to the corresponding dihydronepetalactone, and wild *C. oculata* males collected in early spring contained traces of this dihydronepetalactone. These findings are consistent with the hypothesis that *Chrysopa* males feed on oviparae (the late-season pheromone producing stage of aphids) to obtain nepetalactol as a precursor to iridodial. In the spring, however, wild *C. oculata* males produce less iridodial than do males collected later in the season. Therefore, we further hypothesize that Asian *Chrysopa* eat *A. polygama* to obtain iridoid precursors in order to make their pheromone, and that other iridoid-producing plants elsewhere in the world must be similarly usurped by male *Chrysopa* species to sequester pheromone precursors.

## INTRODUCTION

With ∼6,000 living species, Neuroptera is one of the smaller orders of insects (*Winterton, Hardy & Wiegmann, 2010*), but most larval neuropterans are predacious, often in agricultural systems, lending added importance to this group (*Tauber, Tauber & Albuquerque, 2009*). Green lacewings (Chrysopidae) are the most agriculturally important of the neuropterans because their larvae are generalist predators that actively hunt for aphids, mites, whiteflies, caterpillars, and other small, soft-bodied prey that are common pests on horticultural plants, and in field and tree crops (*McEwen, New & Whittington, 2007*). As do most chrysopid adults, species in the genus *Chrysoperla* feed on nectar and pollen, a characteristic that led to development of artificial diets and mechanized mass rearing of some species (*McEwen, New & Whittington, 2007*; *Nordlund et al., 2001*). All stages of *Chrysoperla* are commercially available for augmentative biological pest control in field and greenhouse crops (*Pappas, Broufas & Koveos, 2011*). In addition, based on volatiles associated with their pollen and nectar consumption, lures for *Chrysoperla* species have been developed to attract wild adults to pest infestations, and to overwintering and egg-laying sites (*Koczor et al., 2014*; *Koczor et al., 2010*; *Tóth et al., 2009*; *Wade et al., 2008*).

The adults of *Chrysopa* species, many of which are naturally important in agricultural systems, are nearly unique among lacewings in that they are predacious, but efforts to develop artificial diets or lures for these species have been largely unsuccessful (*McEwen, New & Whittington, 2007*). Pheromones are potentially useful for attracting generalist predators for augmentative and conservation biological control (*Aldrich, 1999*), and there is ample morphological evidence that in many lacewing species males possess exocrine glands likely to produce aggregation pheromones (*Aldrich & Zhang, 2016*; *Güsten, 1996*). Based upon the meticulous illustrations of male-specific dermal glands in *Chrysopa* (*Principi, 1949*; *Principi, 1954*), we identified the first attractant pheromone for lacewings (*Zhang et al., 2004*). Field-collected males of the goldeneyed lacewing, *Chrysopa oculata* Say, release (1*R*, 2*S*, 5*R*, 8*R*)-iridodial with comparable amounts of nonanal, nonanol and nonanoic acid (*Zhang et al., 2004*). Moreover, iridodial-baited traps attracted *C. oculata* males into traps and females to the vicinity of baited traps (*Chauhan et al., 2007*). Adult *C. oculata* females lack the dermal glands associated with iridodial production, and do not produce iridodial (*Zhang et al., 2004*). Subsequently, we found that the same iridodial stereoisomer as identified from wild *C. oculata* males also attracted adults of *C. nigricornis* Burmeister in the western US (*Zhang et al., 2006a*), and *C. septempunctata* Wesmael in China (*Zhang et al., 2006b*).

The discovery that iridodial powerfully attracted at least three different *Chrysopa* spp., and that the stereochemically correct isomer of iridodial can be prepared using catnip essential oil as starting material (*Chauhan, Zhang & Aldrich, 2004*), encouraged us to pursue pheromone identifications for other lacewings whose males reportedly possess dermal glands similar to those of *Chrysopa* males; i.e., species in the genera *Plesiochrysa*, *Ceratochrysa*, *Nineta*, and *Pseudomallada* (=*Anisochrysa*) (*Aldrich & Zhang, 2016*). But, our plan to pursue pheromone research on some of these chrysopids by rearing them in quarantine was thwarted by the discovery of one us (JRA) that, despite their healthy

appearance, normal fertility and usual amounts of $C_9$ compounds, laboratory-reared *C. oculata* males produced no iridodial. Furthermore, an observation by another of us (Q-HZ) that *C. nigricornis* males caught alive in traps baited with iridodial attempted to eat the lure (unpublished observation), combined with previous reports of *Chrysopa septempunctata* eating the plant known as silver leaf, *Actinidia polygama* (Siebold & Zucc.) Maxim (Actinidiaceae; native to Asia) to obtain the monoterpene iridoids (neomatatabiols) (Fig. S1, compounds 5 and 6) (*Hyeon, Isoe & Sakan, 1968*), suggested that *Chrysopa* males must obtain certain unknown precursors from their diet in order to produce their pheromone.

The objectives of the present study were to (1) devise techniques to feed suspected pheromone precursors to *C. oculata* males and, (2) discover what precursor compound(s) elicit production of iridodial by *C. oculata* males.

## MATERIALS AND METHODS

### Lacewing collection and rearing

Adults of *C. oculata* for the laboratory colony were collected in May of 2008 by sweep net from wild herbaceous vegetation bordering deciduous trees at the Beltsville Agricultural Research Center, Prince George's County, Maryland, USA. Quart wide-mouth Mason® canning jars (Mason Highland Brands, LLC, Hyrum, UT, USA) were used to maintain the adult insects. The jars were positioned horizontally, and nylon organdy cloth (G Street Fabrics, Rockville, MD, USA) was held in place by the screw-top rim used to seal the jars. Jars were provisioned with live parthenogenic pea aphids (Homoptera: Aphidae: *Acyrthosiphon pisum* (Harris)) (supplied by Dr. John Reese, Kansas State University), eggs of the Angoumois grain moth (Gelechiidae: *Sitotroga cerealella* (Oliver); Kunafin "the Insectary," Quemado, TX, USA), and a 10% honey solution. A 5 × 12 cm piece of cardboard was used as a feeding platform. Honey solution was provided in a shell vial (4 ml, 15 × 45 mm; Fisher Scientific, Pittsburgh, PA, USA) with a loose-fitting foam stopper secured at one end of the cardboard with a rubber band. An adhesive strip of a Post-it® paper (50 × 40 mm; 3M, St. Paul, MN, USA) was gently applied to the *Sitotroga* eggs, and the paper was glued (UHUstic®, UHU GmbH & Co., Bühl, Germany) to the other end of the cardboard with the band of moth eggs exposed. The cardboard feeding platform thus prepared was inserted into the bottom of the horizontal jar, and live pea aphid clones (up to several hundred) were added to the cage. Ten to twenty adults could be kept per jar, adding fresh aphids and moth eggs every other day or so, and adding fresh honey solution as needed. In jars used as mating cages (5–10 pairs/jar), a piece of light blue colored paper (providing a color contrast to the green eggs that are laid singly on stalks) was inserted inside the length of the jar as an oviposition substrate. Servicing of these jars was accomplished by working in a cage (30 × 30 × 60 cm; BioQuip Products, Rancho Dominguez, CA, USA) open at one end, and illuminated at the top of the other end by a fluorescent light. Adults from mating jars were moved to new jars weekly, the food platform was removed from the jar with freshly laid lacewing eggs, and the eggs that had been laid were allowed to hatch. Using a camel-hair brush, two first-instar larvae were

transferred to each plastic cup (3/4 oz., snap-on lids; Solo Cup Company, Urbana, IL, USA) with a layer of *Sitotroga* eggs in the bottom. Cups provisioned with only *Sitotroga* eggs were usually sufficient for both larvae to complete all 3 instars and pupate; more than two larvae per cup usually resulted in cannibalism. Lacewing pupae were transferred to the bottom compartment of mosquito breeders (BioQuip Products, Compton, CA, USA) and, upon emergence, the adults were removed from the top compartment. The colony was maintained in an environmental chamber set at 25 °C, 72% relative humidity, and 16:8 h (L:D) photoperiod. Some *C. oculata* males were reared as just described, plus with access to foliage of *Nepeta cataria* (Catnip) (Mountain Valley Seed Inc., Salt Lake City, UT; lot # G2217); some had their antennae removed (antennectomized) 1–5 days after emergence; and some larvae were reared as above, and fed pea aphid clones.

Lacewings are unusual among insects in that adults have chewing mouthparts whereas larvae have piercing/sucking mouthparts (*Tauber, Tauber & Albuquerque, 2009*); therefore, some larvae were reared with methylene blue dye added to the honey solution to verify that larvae ingested materials from the honey water bottles, as did adults. Adult males from these treatments were subsequently chemically sampled and analyzed as described below.

## Scanning election microscopy

Live wild *C. oculata* males were anesthetized with $CO_2$, mounted on copper specimen holders (16 × 29 × 1.5 mm thick) with cryoadhesive, and immersed in liquid $N_2$. The frozen specimens were transferred to an Oxford CT1500 HF cryo-preparation system, and examined using a low temperature scanning electron microscope (LTSEM, Hitachi S-4100; Hitachi, Tokyo, Japan) operated at 2.0 kV (*Erbe et al., 2003*). Micrographs were recorded on Polaroid Type 55 P/N film.

## Chemical standards

$(Z,E)$-Nepetalactone (= (4a$S$,7$S$,7a$R$)-nepetalactone) was prepared from catnip oil; dihydronepetalactone was from hydrogenation of the lactone; $(Z,E)$-nepetalactol (= (1$R$,4a$S$,7$S$,7a$R$)-nepetalactol) was from $NaBH_4$ reduction of the lactone; and 1$R$,2$S$,5$R$,8$R$-iridodial was derived from the $(Z,E)$-nepetalactone as previously described (*Chauhan, Zhang & Aldrich, 2004*). Geranyl and farnesyl pyrophosphates were from Sigma-Aldrich (Saint Louis, MO), as were the following volatile standards (≥95%): geraniol, citronellol, citronellal, linalool, citral, 6-methyl-5-hepten-2-one, 8-hydroxycitronellol, and 8-hydroxycitronellal. $(Z)$-3-Octen-1-ol was from Bedoukian Research, Inc. (Danbury, CT, USA).

## Chemical feeding, extraction of dermal glands, and chemical analysis

Chemical standards were individually fed to adult laboratory-reared *C. oculata* males at 1 μg/μl in the 5% aqueous honey solution for ca. 4 days prior to analysis. For extraction, *C. oculata* adult males were anesthetized with $CO_2$, eviscerated under tap water, the abdominal cuticle (segments 1–8) was removed with microscissors, cleaned of fat under water with micro-forceps, then removed from the water, and dried briefly with tissue paper. Cuticle from a single male was extracted in 10–15 μl of $CH_2Cl_2$ (≥99.9%; Sigma-Aldrich, St. Louis

Missouri, USA) in a Waters Alliance Total Recovery Vial® (deactivated, 12 × 32 mm; Taunton, MA/USA) or the minimum amount of solvent necessary to submerge the cuticles for pooled samples of several males (ca. 50–150 µl) (Zhang et al., 2004). Wild males collected by sweep net, Beltsville MD, May–June, 2008 and 2009, were dissected in like manner the same day as collected.

Gas chromatography (GC) and coinjections were performed in splitless mode using an HP 6890 GC equipped with a DB-5 column (0.25 µm film thickness, 30 m × 0.32 mm ID; J & W Scientific, Folsom, CA). Helium was used as the carrier gas, programming from 50 °C/2 min, to 250 °C at 10 °C/min, then held for 10 min. GC-mass spectrometry (GC-MS) analyses were performed in splitless mode with an electron impact ionization (EI) of 70 eV with an Agilent Technologies 5973 mass selective detector interfaced with 6,890N GC system equipped with either an HP-5MS (30 m × 0.25 mm i.d. × 0.25 µm film) column programmed from 50 °C/2 min, rising to 230 °C at 15 °C/min, then held for 15 min, or using a DB-WaxETR column (0.25 µm film thickness, 30 m × 0.25 mm ID; J & W Scientific, Folsom, CA, USA) programmed at 50 °C/2 min, rising to 230 °C at 15 °C/min, then held for 15 min.

## RESULTS

*Chrysopa* adults are ca. 1.5–2 cm in length, and males are readily attracted to and captured in sticky traps (Fig. 1) (Zhang et al., 2006b). In adult *C. oculata* males the dermal glands (Güsten, 1996) are elliptical (∼12 × 7.5 µm) with a central slit (Fig. 2), and occur on the 3rd–8th abdominal sternites (∼800, 2,100, 2,500, 2,500, 2,300 and 1,500, respectively); corresponding dermal glands are absent in females (Zhang et al., 2004).

Analyses of *C. oculata* revealed that nonanal and nonanol were abundant in extracts of the abdominal sternites of males regardless of whether they were collected in the wild or reared in the laboratory; however, iridodial was absent in extracts of laboratory-reared *C. oculata* males (Figs. 3A and 3B; Table 1). Rearing *C. oculata* males in isolation from conspecific males did not result in production of iridodial (Table 1), and removing the antennae of *C. oculata* males had no affect on production of iridodial (Fig. S2). Access of *C. oculata* males to *Nepeta cataria* (catnip) foliage in the laboratory did result in a detectable level of iridodial (Fig. S3); however, this level was far below that seen for wild *C. oculata* males (Table 1). In wild males collected by sweep netting foliage in early spring (i.e., not from iridodial-baited traps) the mean iridodial percentage relative to the abundances of nonanal and nonanol was 14.30 % (±SEM = 3.72) (Table 1). Analysis of one male caught in an iridodial-baited trap (14 May 2008, Beltsville, MD, USA) to which the captured males had access to the lure, showed that this male produced much more iridodial (40.71 %) than the normal mean abundance of iridodial in wild *C. oculata* males (Table 1). (*Z*)-3-Octen-1-ol was used as an internal standard to quantitate pheromone production per wild *C. oculata* males collected in May 2008; extracts of single males contained 20.42 ± 6.88 ng iridodial/male (mean ± SEM; N = 8) (Data S8).

Feeding naturally common monoterpene alcohols and aldehydes to *C. oculata* males did not stimulate production of iridodial (Table 2, experiment numbers 1–8). However,
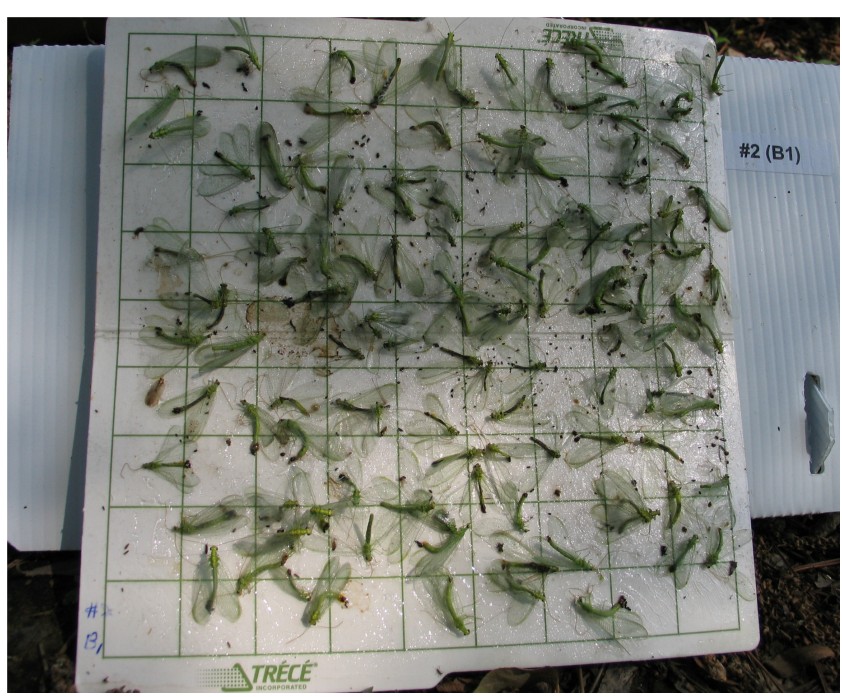

**Figure 1**  **Male *Chrysopa septempunctata* captured in pheromone-baited trap, Shengyang, China** (*Zhang et al., 2006a*; *Zhang et al., 2006b*). *Chrysopa* females come to the vicinity of iridodial-baited traps, but are seldom caught (*Chauhan et al., 2007*).

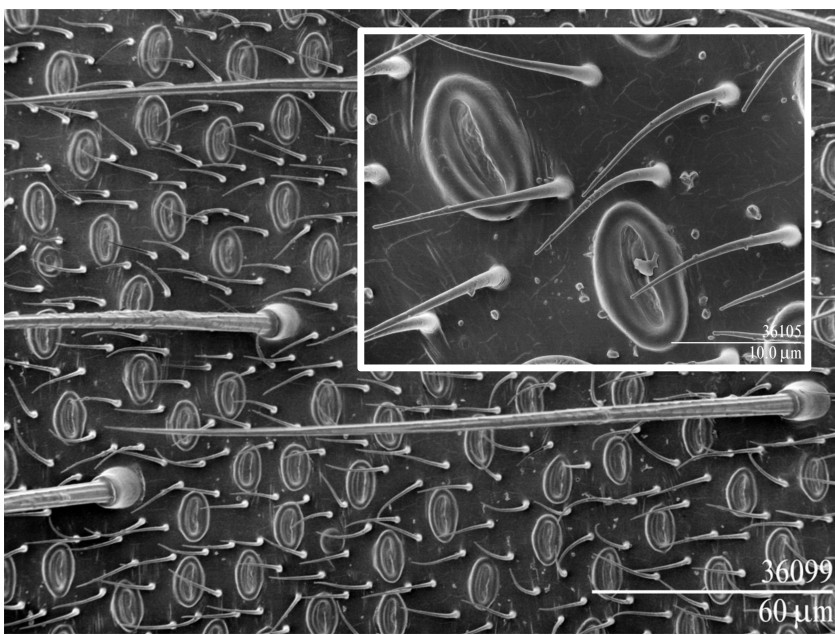

**Figure 2**  **Scanning electron micrographs of the male-specific dermal glands of *Chrysopa oculata*.**  Low temperature scan (*Erbe et al., 2003*) with insert showing close-up of two dermal glands.

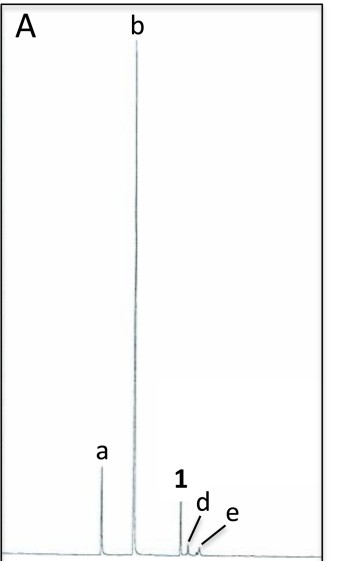
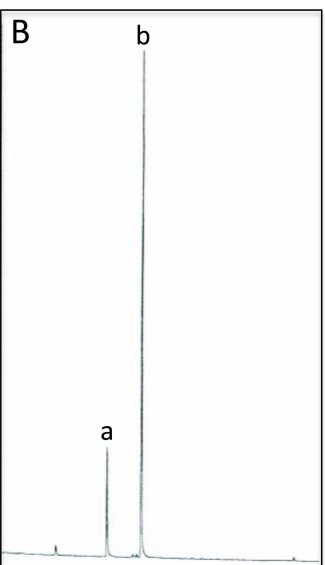
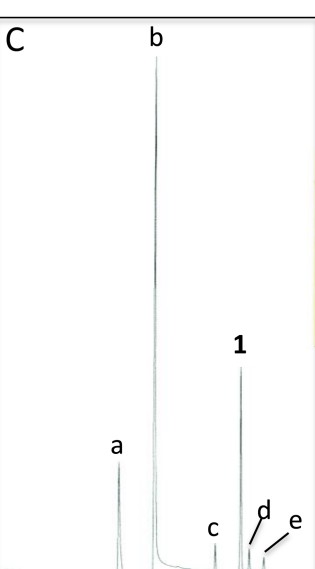

**Figure 3   Total ion chromatograms of abdominal cuticular extracts of male *Chrysopa oculata*.**
(A) Field-collected, (B) laboratory-reared and, (C) laboratory-reared fed (1*R*,4*S*,4a*R*,7*S*,7a*R*)-dihydronepetalactol (see Table 2). (Column, 30 m DB-WAXetr: **a**, nonanal; **b**, nonanol; **c**, (*Z*)-4-tridecene; **1**, (1*R*,2*S*,5*R*,8*R*)-iridodial; **d** & **e**, 168 MW isomers.)

this series of feeding trials did reveal that males evidently possess reductase and saturase enzymes capable of reducing aldehydes to alcohols, and of saturating double bonds in these molecules. These reactions were essentially unidirectional; for example, geranial was completely converted to geraniol (Table 2, experiment number 2), whereas geraniol was only slightly isomerized to nerol but aldehydes were not produced (Table 2, experiment number 6). Furthermore, the abundances of $C_9$ compounds were not affected; nonanal, nonanol and nonanoic acid occurred in ratios within their ranges for wild-caught males for all experiments shown in Table 2.

Feeding male goldeneyed lacewings the common aphid pheromone components, (4a*S*,7*S*,7a*R*)-nepetalactone and (1*R*,4a*S*,7*S*,7a*R*)-nepetalactol, produced positive results. While feeding nepetalactone did not result in production of iridodial, about 75% of this lactone was converted to the dihydronepetalactone (Table 2, experiment number 9). Furthermore, dihydronepetalactone was detected at low, but unequivocal levels in some samples from wild *C. oculata* males (Figs. S4 and S5). *Chrysopa oculata* males fed (1*R*,4a*S*,7*S*,7a*R*)-nepetalactol converted this compound to (1*R*,2*S*,5*R*,8*R*)-iridodial (82.7%; Table 2, experiment number 10; Fig. 3C), with two later eluting 168 MW compounds accounting for 17.3% of the other newly appearing components, as well as (*Z*)-4-tridecene from the defensive prothoracic glands (Fig. 3C, compound **c**) (*Aldrich et al., 2009*). Two additional feeding experiments were conducted as for experiment 10 (Table 2); one of these experiments using the same GC-MS conditions ($N = 9$ males) showed 54.90% conversion to (1*R*,2*S*,5*R*,8*R*)-iridodial with the two later eluting 168 MW components totaling 40.10%, and the second experiment ($N = 4$ males) analyzed

**Table 1  Volatiles from abdominal cuticle of field-collected and laboratory-reared *Chrysopa oculata* males.** Wild *C. oculata* males were collected by sweep net, Beltsville, Maryland, and *C. oculata* laboratory-reared males (see text for details) were sampled for comparisons. One *C. oculata* male was from a field trap baited with a lure including synthetic iridodial. Abdominal cuticle (segments 3–8) for chemical analyses were prepared as described previously (*Zhang et al., 2004*).

| Source/date | N[a] | Compound (%) | | | %∑[c] |
|---|---|---|---|---|---|
| | | Nonanal | Nonanol | Iridodial[b] | |
| Field/14 May 2009 | 4 | 13.06 | 80.68 | 2.35 | 96.09 |
| Field/18 May 2009 | 2 | 15.81 | 80.16 | 2.12 | 98.09 |
| Field/22 May 2009 | 1 | 10.31 | 42.01 | 38.13 | 90.45 |
| Field/28 May 2009 | 1 | 30.09 | 50.06 | 16.11 | 96.26 |
| Field/28 May 2009 | 1 | 13.56 | 67.55 | 16.19 | 97.30 |
| Field/28 May 2009 | 1 | 8.84 | 74.88 | 14.06 | 97.78 |
| Field/1 June 2009 | 1 | 32.24 | 54.82 | 9.94 | 97.00 |
| Field/1 June 2009 | 1 | 13.69 | 65.20 | 15.53 | 94.42 |
| **Mean:** | | **13.95** | **64.42** | **14.30** | **95.92** |
| **±SEM:** | | **3.81** | **4.73** | **3.72** | |
| Field trap[d]/13 May 2008 | 1 | 16.43 | 38.93 | 40.71 | 96.07 |
| Lab/27 June 2008[e] | 8 | 21.28 | 76.26 | 0 | 97.54 |
| Lab/13 Aug 2008[e] | 5 | 21.37 | 69.34 | 0 | 90.71 |
| Lab/24 Nov 2008[e] | 6 | 11.20 | 86.12 | 0 | 97.32 |
| Lab/24 Nov 2008[e] | 7 | 18.60 | 75.74 | 0 | 94.34 |
| Lab/5 Jan 2009[f] | 5 | 16.58 | 79.42 | 0 | 96.00 |
| **Mean:** | | **17.81** | **77.38** | **0** | **95.18** |
| **±SEM:** | | **1.88** | **1.73** | | |

**Notes.**
[a] In samples where $N > 1$, multiple males were pooled and analyzed as a single sample by GC-MS on a 30 m DB-WaxETR column.
[b] (1R,2S,5R,8R)-Iridodial (*Chauhan, Zhang & Aldrich, 2004*).
[c] Percentage of total volatiles; nonanoic acid (poorly resolved chromatographically) accounted for the majority of non-included volatiles.
[d] This *C. oculata* male was collected in a trap baited with 5 mg of iridodial plus 1 mg of skatole per 50 μl of octane to the well of gray rubber septa (5-mm sleeve-type, The West Co., Lititz, PA); the trap used was as previously described (*Zhang et al., 2004*), and it was deployed at the Agricultural Research Center-West, B Beltsville, MD.
[e] Reared singly as adults.
[f] Reared in a group as adults.

using a 30m HP-5 column resulted in 100% conversion to (1R,2S,5R,8R)-iridodial (Data S7, #10a & b).

Finally, feeding experiments conducted with *C. oculata* larvae failed to stimulate more than trace levels of iridodial in the resulting male adults. Providing pea aphid clones to larvae during rearing yielded at most only traces of iridodial in the ensuing adult males (Data S11). While *Chrysopa oculata* larvae provided with honey water solution containing methylene blue turned decidedly blue, verifying this method as an appropriate means to feed suspected pheromone precursors to larvae, feeding geranyl or farnesyl pyrophosphates did not stimulate detectable production of iridodial in the ensuing adult males (Data S11). Feeding *C. oculata* larvae with (1R,4aS,7S,7aR)-nepetalactol, which in laboratory-reared adult males resulted in wild-type levels of (1R,2S,5R,8R)-iridodial, produced trace levels of iridodial far lower than wild-type levels of the pheromone (Data S10).

## DISCUSSION

Coincidence of male-specific dermal glands with extraction of (1R,2S,5R,8R)-iridodial from the 3rd–8th abdominal sternites strongly implicates these glands as the pheromone

**Table 2 Compounds produced by laboratory-reared *Chrysopa oculata* males fed various exogenous terpenoids.** Sampling and rearing methods described in text; 1 μg/μl test compound in honey water, analyzed by gas chromatography-mass spectrometry using a 30 m DB-WaxETR column.

| No. | N[a] | Compound fed[b] | Compound(s) produced from treatment (%)[c] | | | |
|---|---|---|---|---|---|---|
| | | | a | b | c | d |
| 1 | 8 | | (16) | (3.3) | (9.7) | (71) |
| 2 | 12 | | (9.9) | (8.3) | (42.3) | (39.5) |
| 3 | 9 | | (100) | | | |
| 4 | 10 | | (100) | | | |
| 5 | 7 | | (95.3) | (4.7) | | |
| 6 | 5 | | (4.3) | (95.7) | | |
| 7 | 15 | | (100) | | | |
| 8 | 15 | | (100) | | | |
| 9 | 12 | | (23.3) | (76.7) | | |

**Table 2** (*continued*)

| No. | N[a] | Compound fed[b] | Compound(s) produced from treatment (%)[c] | | | |
|---|---|---|---|---|---|---|
| | | | a | b | c | d |
| 10 | 10 | (82.7) | | | | |

**Notes.**

[a] Number of males pooled for analysis.

[b] Sources of standards listed in text; (1) 3,7-dimethyl-1,6-octadien-3-ol (linalool), (2) (*Z*/*E*)-3,7-dimethyl-2,6-octadienal (citral: 43% *Z*-isomer, neral + 57% *E*-isomer, geranial), (3) 6-methyl-5-hepten-2-one, (4) 2,6-dimethyl-5-heptenal (citronellal), (5) 2,6-dimethyl-5-heptenol (citronellol), (6) (*E*)-3,7-dimethyl-2,6-octadien-1-ol (geraniol), (7) (*E*)-3,7-dimethyl-8-hydroxy-6-octen-1-al (8-hydroxycitronellal), (8) (*E*)-2,6-dimethyloct-2-ene-1,8-diol (8-hydroxycitronellol), (9) (4a*S*,7*S*,7a*R*)-nepetalactone and, (10) (1*R*,4*S*,4a*R*,7*S*,7a*R*)-dihydronepetalactol. Purities of all standards (except for iridodial) were ≥95%; synthetic and natural iridodial analyzed by GC existed with two later eluting 168 MW isomers (Fig. 3; compounds d and e), here accounting for 10.2% and 7.1%, respectively, of the 168 MW compounds.

[c] Abdominal cuticle (segments 3–8) for chemical analyses of *C. oculata* male-produced volatiles were prepared as described previously (*Zhang et al., 2004*). Compounds produced from fed precursors for which synthetic standards were available were verified by coinjections: (2c & 6a) nerol; (2d, 5b & 6b) geraniol; (4a & 5a) citronellol; (9a) (4a*S*,7*S*,7a*R*)-nepetalactone; (9b) (4a*S*,7*S*,7a*R*)-dihydronepetalactone and, (10a) (1*R*,2*S*,5*R*,8*R*)-iridodial. Other compounds were tentatively identified by near matches to mass spectra of compounds in the National Institute of Standards and Technology (NIST) mass spectral library: (1a) 3,7-dimethyl-6-octen-3-ol (1,2-dihydrolinalool); (1b) (*Z*)-3,7-dimethyl-2,6-octadien-1-ol; (1c) 2,6-dimethyl-7-octene-2,6-diol; (1d) (*E*)-2,6-dimethyl-2,7-octadiene-1,6-diol; (2a & 3a) 6-methyl-5-hepten-2-ol; (2b) 3,7-dimethyl-6-octen-1-ol. Compound 7a and 8a yielded a less than a perfect match for 3,7-dimethyl-1,7-octanediol; based upon previously seen glandular reactions, this compound is likely 2,6-dimethyl-1,8-octanediol.

source (*Zhang et al., 2004*). Surprisingly, only males are caught in traps baited with this iridodial (*Zhang et al., 2004*; *Zhang et al., 2006a*; *Zhang et al., 2006b*); however, females are drawn to the vicinity of, but seldom enter, iridodial-baited traps (*Chauhan et al., 2007*). Presumably, females stop short of entering traps because the close-range substrate-borne vibrational signals to which females are ultimately attracted (*Henry, 1982*) are disrupted by trapping males. The $C_9$ compounds are unattractive to *C. oculata*, quantitatively much less variable than iridodial, and inhibitory to iridodial attraction, suggesting these compounds play a role independent from that of iridodial (*Zhang et al., 2004*).

Previous laboratory rearing studies with *Chrysopa oculata* showed that males produced fertile matings when fed only sugar and water, whereas females needed to feed on pea aphid clones in order to mate and produce fertile eggs (*Tauber & Tauber, 1973*). Our results support these finding, but also make it clear that *C. oculata* males are unable to make pheromone on this feeding regimen. Iridodial production in *C. oculata* males was not stimulated by (1) antennectomy of sexually mature *C. oculata* males, which in some group-reared insects stimulates pheromone production (e.g., *Dickens et al., 2002*); (2) providing access to catnip plants, *Nepeta cataria*, containing the nepetalactone aphid pheromone component (*Pickett, Allemann & Birkett, 2013*) or; (3) rearing *C. oculata* males in isolation, which in some insects is required for maximal pheromone production (*Ho et al., 2005*; *Khrimian et al., 2014*).

Cyclopentanoid natural products based on an iridoid structure are widespread in plants and insects (*Hilgraf et al., 2012*; *Lorenz, Boland & Dettner, 1993*), and incorporation of ($^{14}$C) mevalonolactone by the stick insect, *Anisomorpha buprestoides* (Stoll) (Phasmatodea:

Pseudophasmatidae), and the catnip plant (*N. cataria*) demonstrated that biosynthesis of their respective iridoids, anisomorphal and nepetalactone, proceed via parallel terpene pathways from acyclic precursors, particularly geraniol (*Meinwald et al., 1966*). Larvae of leaf beetles (Coleoptera: Chrysomelidae) from four different genera showed that biosynthesis of the iridoid defensive compound, chrysomelidial, proceeds from geraniol via an $\omega$-oxidation sequence to 8-hydroxygeraniol, with the eventual cyclization of 8-oxocitral to form the characteristic iridoid cyclopentanoid ring structure (*Hilgraf et al., 2012*; *Lorenz, Boland & Dettner, 1993*; *Veith et al., 1994*). Certain rove beetles (Coleoptera: Staphylinidae: *Philonthus* spp.) also produce defensive secretions containing iridoids (e.g., plagiodial), but unlike enzymes from iridoid-producing leaf beetle larvae, the *Philonthus* enzyme is able to oxidize and cyclize saturated substrates such as citronellol (*Weibel et al., 2001*). In plants, including a catnip species (*N. racemosa*) (*Hallahan et al., 1995*), the cyclization reactions to iridoids proceed via 10-hydroxygeraniol and 10-oxogeranial rather than 8-hydroxygeraniol/al (*Geu-Flores et al., 2012*). Furthermore, *Hilgraf et al. (2012)* stressed that there are still many open questions concerning the biosynthesis of iridoids, particularly "saturated" iridoids such as iridodial.

In contrast to other iridoid-producing insects and plants whose biosynthetic pathways have been investigated, *Chrysopa* males are evidently incapable of cyclizing geraniol or other acyclic analogs to form the cyclopentanoid ring structure characteristic of iridoid compounds. Thus, feeding acyclic monoterpene alcohols and aldehydes to *C. oculata* males did not stimulate production of iridodial. However, our feeding trials revealed that *C. oculata* males are capable of reducing aldehydes to alcohols and of saturating double bonds. Moreover, males fed the common aphid pheromone component, (4a*S*,7*S*,7a*R*)-nepetalactone, converted ∼75% to dihydronepetalactone, and males fed the other common aphid pheromone component, (1*R*,4a*S*,7*S*,7a*R*)-nepetalactol, converted this bicyclic iridoid to (1*R*,2*S*,5*R*,8*R*)-iridodial. Interestingly, analyses of wild *C. oculata* males collected in May often revealed the presence of dihydronepetalactone.

One interpretation of these data is that *C. oculata* males must eat aphid oviparae to obtain nepetalactol in order to make their pheromone. Indeed, in northern California the peak late-season attraction of green lacewings to nepetalactol (nepetalactone is unattractive) occurs at least a month earlier than the peak in aphid oviparae (*Symmes, 2012*), consistent with the hypothesis that *Chrysopa* males feed on oviparae to obtain nepetalactol as a precursor for iridodial. These dynamics indicate there is sufficient time for *Chrysopa* males to feed on oviparae, produce iridodial, mate, and have conspecific females' offspring reach the prepupal overwintering stage (*Uddin, Holliday & MacKay, 2005*). However, adult males from laboratory-reared *C. oculata* larvae fed nepetalactol still failed to produce wild-type levels of iridodial even though wild *C. oculata* males collected early in the spring produce less iridodial than do males collected later in the season (*Zhang et al., 2004*). Although some aphids produce oviparae under stressed conditions in summer (*Hardie, 1985*), it seems unlikely that these oviparae are a reliable or abundant enough source to sustain *Chrysopa* male pheromone production. Therefore, we further hypothesize the *raison d'être* that Asian *Chrysopa* eat fruit and foliage of silver leaf (*A. polygama*) is to obtain iridoid precursors necessary to make their pheromone; we believe that other iridoid-producing

plants (e.g., *Hilgraf et al., 2012*; *Prota et al., 2014*) elsewhere in the world must be similarly usurped by male *Chrysopa* species to sequester iridoid pheromone precursors.

Contrary to *Chrysoperla* green lacewings whose adults are not predacious, *Chrysopa* spp. lacewing adults are predacious (*Tauber, Tauber & Albuquerque, 2009*), and appear to exhibit pharmacophagy; that is, they "search for certain secondary plant substances directly, take them up, and utilize them for a specific purpose other than primary metabolism" (*Boppré, 1984*). A prime example of pharmacophagy are male *Bactrocera* fruit flies (Tephritidae) that feed on plants to obtain their pheromone precursor, methyl eugenol (*Tan & Nishida, 2012*). Indeed, males of certain lacewings (i.e., *Ankylopteryx exquisite* (Nakahara) (*Pai et al., 2004*), and *Mallada basalis* (Walker) (*Oswald, 2015*; *Suda & Cunningham, 1970*)) are also powerfully attracted to methyl eugenol for unknown reasons (*Tan & Nishida, 2012*). In addition, certain chrysomelid beetle larvae discharge iridoid allomones that may be synthesized *de novo*, which is considered ancestral, or produced via the more evolutionarily advanced mechanism, sequestration from plants (*Kunert et al., 2008*). Increasingly, pharmacophagy is being recognized as a widespread phenomenon in insects, and *Wyatt (2014)* has extended the concept of pharmacophagy to include molecules produced by bacteria that are used as pheromones, such as locust phase-change pheromones produced by gut bacteria. If male *Chrysopa* spp. lacewings actually do seek out aphid oviparae to obtain nepetalactol as a precursor to iridodial, and in this regard it should be noted that only *Chrysopa* males are attracted to nepetalactol (*Koczor et al., 2015*), then the concept of pharmacophagy must be further extended to include this type of predator/prey interaction. Whether or not sequestration of iridodial precursors from oviparae and/or iridoid-containing plants is truly the explanation for lack of pheromone in laboratory-reared *Chrysopa* awaits further research.

## CONCLUSIONS

Goldeneyed lacewing males, *Chrysopa oculata* (Neuroptera: Chrysopidae), produce (1*R*, 2*S*, 5*R*, 8*R*)-iridodial as an aggregation pheromone from specialized dermal glands on the abdomen; however, seemingly normal laboratory-reared males of *C. oculata* do not produce iridodial. Feeding studies with *C. oculata* showed that males of these predatory insects fed one of the common aphid sex pheromone components, (1*R*, 4a*S*, 7*S*, 7a*R*)-nepetalactol, sequester this compound and convert it to the stereochemically correct lacewing pheromone isomer of iridodial. These data, combined with literature accounts of other *Chrysopa* species from the Oriental region that feed on iridoid-producing plants, suggest these (and some other) lacewing species must obtain precursors from aphid oviparae and/or certain plants containing iridoids in order to make pheromone. The phenomenon, known as pharmacophagy, whereby an insect searches for certain secondary plant substances and sequesters the chemicals for a specific purpose other than primary metabolism, is widespread among phytophagous insects but, to our knowledge, is unknown among lacewings or other predacious insects. Our findings, if verified, have significant implications for lacewing-based biological control of aphids and other small arthropod pests. In particular, it would be highly desirable to test the pharmacophagy hypothesis in a more

natural context by feeding aphid oviparae to *Chrysopa* males. Additionally, metabolic assays using deuterium labeled suspected precursors to iridodial would more precisely substantiate the biosynthetic pathway to iridodial in *Chrysopa* males.

## ACKNOWLEDGEMENTS

The authors wish to dedicate this manuscript to the memory of Dr. Murray S. Blum (July 19, 1929–March 22, 2015), University of Georgia, Athens, who was a true pioneer of chemical ecology and mentor to many of us in this field. On the occasion of her 100th birthday this year, we recognize the *grande dame* of neuropterists, Maria Matilde Principi, for her inspirational work describing the pheromone glands of lacewings. We are also grateful to Professor Wilhelm Boland (Department of Bioorganic Chemistry, Max Planck Institute for Chemical Ecology, Jena, Germany) for helpful discussions. Dr. John Reese (Department of Entomology, Kansas State University, Manhattan) provided live pea aphids weekly for the duration, which made this research possible. Finally, one of us (JRA) thanks Mr. Ed Clark for his expert technical assistance.

### Funding

Most of the research was performed at the USDA-ARS laboratory in Beltsville, MD, when Dr. Aldrich was a Research Entomologist with this agency. Dr. Qing-He Zhang was a postdoctoral Research Associate with Dr. Aldrich at this time ($\sim$2002–2003). The USDA-ARS provided funding in the form of salaries, and laboratory space and supplies, but funders did not have a role in the research other than broad National Program guidelines. The funders had no role in study design, data collection and analysis, decision to publish, or preparation of the manuscript.

### Grant Disclosures

The following grant information was disclosed by the authors:
USDA-ARS laboratory in Beltsville.

### Competing Interests

Dr. Qing-He Zhang is an employee of Sterling International, Inc., Spokane, WA. Dr. Jeffrey R. Aldrich is the founder and sole employee of Jeffrey R. Aldrich consulting, LLC.

### Author Contributions

- Jeffrey R. Aldrich conceived and designed the experiments, performed the experiments, analyzed the data, wrote the paper, prepared figures and/or tables.
- Kamal Chauhan contributed reagents/materials/analysis tools, reviewed drafts of the paper.
- Qing-He Zhang analyzed the data, reviewed drafts of the paper.

### Data Availability

   Raw data can be found in the Supplemental Information.

## Supplemental Information

Supplemental information for this article can be found online at http://dx.doi.org/10.7717/peerj.1564#supplemental-information.

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
