# Peer review of "Pharmacophagy in green lacewings (Neuroptera: Chrysopidae: Chrysopa spp.)?"

_PeerJ, doi:10.7717/peerj.1564_

## Round 0.1 · original submission · Major Revisions

· Academic Editor

Major Revisions

Dear Authors,

Your MS is now reviewed by 3 independent experts. While the manuscript is of general interest, I believe that there are some additional experiments and extensive rewriting is needed before the MS can be finally accepted. In general, there is concern/suggestions about the detection of metabolic intermediates and shading light on the bio-chemical pathway involved in such. I personally believe that addition of such points will highly improve your manuscript quality.

·

Basic reporting

It is an interesting piece of work and the findings were represented with great clarity in the language and with the help of tables and figures. The article needs some minor corrections in the text and a few clarifications. One of the reference mentioned in text is missing in the bibliography.

Experimental design

The experiments were carefully designed to address the hypothesis. But sample processing for GC-MS analysis and MS programming information should be shared in detail.

Validity of the findings

No comments

Comments for the author

No comments

Reviewer 2 ·

Basic reporting

See below

Experimental design

See below

Validity of the findings

See below

Comments for the author

Basic observation:

The manuscript entitled with “Pharmacophagy in green lacewings (Neuroptera: Chrysopidae: Chrysopa spp.)” submitted by Aldrich et al. is a good manuscript with potential, but not satisfactory at this stage due to several problems. In this manuscript the authors have attempted to investigate/compare the endogenous production of a pheromone in male in wild type and laboratory conditions. The findings are of interest, but lacks proper controls and references and further experimental data is required to justify their conclusions.

Here are some of the general and specific comments:

a) Abstract is not well written and lengthy. It need to be shortened and made precise.
b) The exact objective/s of the study is not clear and need to be mentioned in the light of proper citations.
c) The application side of this study (potential pest control etc.) need to be discussed in the context properly.
d) Introduction section is poorly written and it needs extensive modification.
e) The organization of results section need editing and rearrangement. The figures must be organized from bigger to smaller scale such as animal, gland and extracts and small compounds.

Specific comments:

a) In some context, the authors claim is without based on any specific experimental data. The authors have discussed about the “chemical reaction” but have not provided any evidence in support of that. To prove this idea more experimental evidences are required.
b) To understand the basis of chemical structure determined, the NMR/MALDI-MS data will be required.
c) The authors should “confirm” their claim by using a proper “metabolic labelling” experiment where a precursor with radio-labelled/spin-labelled isotope. Such a probe can be added in the food and production of the derivatives in laboratory conditions should be analysed.
d) The possible bio-chemical pathway involved in such process of Pheromone production should be indicated and discussed in details.

Reviewer 3 ·

Basic reporting

.

Experimental design

.

Validity of the findings

.

Comments for the author

This paper reports the Goldeneyed lacewing males, Chrysopa oculata (Neuroptera: Chrysopidae), which produces (1R,2S,5R,8R)-iridodial as
an aggregate from specialized dermal glands on the abdomen. It was also shown that seemingly normal laboratory-reared males of C. oculata do not produce iridodial. Feeding studies with C. oculata further demonstrate that males of these predatory insects fed one of the common aphid sex pheromone components, (1R,4aS,7S,7aR)- nepetalactol, sequester this compound and convert it to the stereochemically correct lacewing pheromone isomer of iridodial. The experimental designs were mostly carried out correctly. This present manuscript fits well Peer J readership’s interests and may become suitable for publications, however, few more additional experiments can covert this manuscript a much better one.
The authors should provide quantitative and qualitative data on the pheromone produced by individual animals and also a time point distribution (how much is produced in each day). Identifying the metabolic intermediates is also important.

---

## Round 0.2 · accepted · Accept

· Academic Editor

Accept

Dear Authors,

I have gone through your line-by-line response and the original comments carefully. I noted that extensive editing is done and the quality of the manuscript has been improved. Based on the reviewer's comment and personal cross checking of the revised version, I am happy to inform you that your manuscript has been considered to be acceptable.

However, at this juncture, I would like to request you to add a small paragraph to the discussion section regarding the "future prospect of this study". You may also self declare some of the shortcomings which could not be accommodated in this manuscript such as metabolic labeling and characterization of the metabolites by MALDI-MS. At least pointing these aspects will really make this manuscript a perfect one. Please pay attention on the language, grimmer and spellings at this stage and if possible, by a native English speaker.

I congratulate once again to you for considering PeerJ as a suitable journal for your manuscript.

·

Basic reporting

No comments

Experimental design

No comments

Validity of the findings

No comments

---

## Author Rebuttal · Round 0.2

# Jeffrey R. Aldrich consulting, LLC

7 November 2015

Dear PeerJ Editors:

We have extensively revised our manuscript, "Pharmacophagy in green lacewings (Neuroptera: Chrysopidae: Chrysopa spp.)?", in accordance with the constructive critiques provided by the reviewers. The revised manuscript is undoubtedly much improved, and we are hopeful that the editors now see fit to expeditiously accept our manuscript for publication.

The authors thank all PeerJ editors involved in making our first PeerJ experience a positive one!

Sincerely,

Ph.D. Research Entomologist / Chemical Ecologist

519 Washington Street
Santa Cruz, CA / USA 95060
Phone: (301503-8288
E-Mail: DrJeffAldrich@gmail.com

Re: rebuttal & revision of Aldrich et al., "Pharmacophagy in green lacewings (Neuroptera: Chrysopidae: Chrysopa spp.)?"

**Editor's comments:**
**While the manuscript is of general interest, I believe that there are some additional experiments and extensive rewriting is needed before the MS can be finally accepted. In general, there is concern/suggestions about the detection of metabolic intermediates and shading light on the bio-chemical pathway involved in such. I personally believe that addition of such points will highly improve your manuscript quality.**

**Response:** We have extensively revised the manuscript according to reviewers' suggestions, and included raw data files substantiating all our results. In the process we have moved introductory remarks regarding Maria Principi to acknowledgements, added references and discussion of the biochemical pathways to iridoids in other insects, added new data on quantitation of pheromone production per male, and reordered the results and other sections of the presentation as suggested by Reviewer 2.

**Reviewer 1 (Apratim Maity)**

**Basic reporting:**
**It is an interesting piece of work and the findings were represented with great clarity in the language and with the help of tables and figures. The article needs some minor corrections in the text and a few clarifications. One of the references mentioned in text is missing in the bibliography. The experiments were carefully designed to address the hypothesis. But sample processing for GCMS analysis and MS programming information should be shared in detail.**

**Annotated manuscript comments/questions followed by authors' response:**

**1) Pages 1, 4, 12 & 14 of PeerJ Preprint: Questioning whether "(1$R$,4a$S$,7$S$,7a$R$)-nepetalactol" should be the "dihydronepetalactol."**

**Response:** "(1$R$,4a$S$,7$S$,7a$R$)-nepetalactol" is the correct compound; *i.e.* this is one of the two common aphid pheromone components.

**2) Pg. 6: Asked in what context "Supplemental Figure 1, compounds 5 and 6" is being cited?**

**Response:** This citation was rearranged slightly to make it clear of the context: "…to obtain the monoterpene iridoids (neomatatabiols) (Supplemental Figure 1, compounds **5** and **6**)…"

**3) Pg. 6: Chauhan et al. (2004) is not listed in the literature cited.**

**Response:** Good catch; Chauhan et al. (2004) has been added to the reference list.

**4) Pg. 10: Sample processing and method of injection is to be mentioned for GC-MS analysis. Mention details of MS programming also.**

**Response:** Much more detail for sample preparation (in addition to citing Zhang et al. 2004) was added in the section on "Chemical feeding, extraction of dermal glands, and chemical analysis." Also, we added "splitless mode" as the descriptor for GC-MS analysis conditions.

**5) Pg. 11: Data for the male caught in one iridodial-baited trap (14 May 2008, Beltsville, MD) to which the captured males had access to the lure is not included in Table 1.**

**Response:** These data were added to Table 1 (along with the raw data file in supplemental material) as a "Field-Trap" line between "Field" and "Lab" data.

**6) Pg. 11: In Table 2, #4 refers to citronellal/ol, not geraniol/al.**

**Response:** Good catch by the reviewer! The citation has been changed to "#2", which is the experiment involving geraniol feeding.

**Reviewer 2 (Anonymous)**

**Basic observation:**
**The manuscript entitled with "Pharmacophagy in green lacewings (Neuroptera: Chrysopidae: Chrysopa spp.)" submitted by Aldrich et al. is a good manuscript with potential, but not satisfactory at this stage due to several problems. In this manuscript the authors have attempted to investigate/compare the endogenous production of a pheromone in male in wild type and laboratory conditions. The findings are of interest, but lacks proper controls and references and further experimental data is required to justify their conclusions.**

**Response:** Reviewer 2 has made the most critical comments, most of which have been incorporated into to our revision (as detailed below), undoubtedly resulting in an improved, and more thorough narrative and interpretation of results.

**General comments:**

**1) Abstract is not well written and lengthy. It need to be shortened and made precise.**

**Response:** The abstract has been rewritten for clarity and to reduce the length; the word count has been reduced from 383 to 311.

**2) The exact objective/s of the study is not clear and need to be mentioned in the light of proper citations.**

**Response:** A specific objectives statement has been added to the end of the introduction:
"The objectives of the present study were to 1) devise techniques to feed suspected pheromone precursors to *C. oculata* males and, 2) discover what precursor compound(s) elicit production of iridodial by *C. oculata* males."

**3) The application side of this study (potential pest control etc.) need to be discussed in the context properly.**

**Response:** The introductory remarks regarding Maria Principi, which were admittedly distracting, have been moved to Acknowledgements. An extensive addition to the Introduction has been added (with the addition of several new references) to describe the practical significance of lacewings and of our study as follows:
"Green lacewings (Chrysopidae) are the most agriculturally important of the neuropterans because their larvae are generalist predators that actively hunt for aphids, mites, whiteflies, caterpillars, and other small, soft-bodied prey that are common pests on horticultural plants, and in field and tree crops (McEwen et al. 2007). While most chrysopids are also predacious as adults, species in the genus *Chrysoperla* feed on nectar and pollen, a characteristic that led to development of artificial diets and mechanized mass rearing of some species (McEwen et al. 2007; Nordlund et al. 2001). All stages of *Chrysoperla* are commercially available for augmentative biological pest control in field and greenhouse crops (Pappas et al. 2011). In addition, based on volatiles associated with their pollen and nectar consumption, lures for *Chrysoperla* species have been developed to attract wild adults to pest infestations, and to overwintering and egg-laying sites (Koczor et al. 2014; Koczor et al. 2010; Tóth et al. 2009; Wade et al. 2008).
Many other lacewings whose adults are predacious are naturally important in agricultural systems, most notably *Chrysopa* species, but efforts to develop artificial diets or lures for these species have been largely unsuccessful (McEwen et al. 2007). Pheromones are potentially useful for attracting generalist predators for augmentative and conservation biological control (Aldrich 1999), and there is ample morphological evidence that in many lacewing species males possess exocrine glands likely to produce aggregation pheromones (Aldrich and Zhang 2016; Güsten 1996)…"

**4) Introduction section is poorly written and it needs extensive modification.**

**Response:** The Introduction has been extensively rewritten with a clear statement of objectives at the end, and the addition of narrative and references clarifying the agricultural importance of lacewings and the significance of our study (as described above). In addition, we have more clearly introduced the idea of using plant volatiles and pheromones as a means to manipulate lacewings for enhanced biological pest control, and added a list of genera for which male-specific dermal glands likely to produce pheromones have been found, referencing our forthcoming Annual Review of Entomology manuscript (Spring of 2016) on "Chemical Ecology of Neuroptera."

**5) The organization of results section need editing and rearrangement. The figures must be organized from bigger to smaller scale such as animal, gland and extracts and small compounds.**

**Response:** The suggested rearrangements have been made in the ordering of the Results, with corresponding reordering of the Methods and Materials section, as follows:
a) The insect collection and rearing (with much added new detail) is presented first, with Figure 1 showing the whole insects on a sticky trap baited with iridodial.
b) The scanning electron micrograph of the male-specific dermal glands is shown in Figure 2.
c) Figure 3 then shows gas chromatograms of volatiles from a wild *C. oculata* male (A), a laboratory-reared male (B), and a laboratory-reared male fed the aphid pheromone compound, nepetalactol (C). Table 1 presents more extensive data (with corresponding raw data supplemental files) on analyses of wild versus laboratory-reared *C. oculata* males; and Table 2 presents the chemical structures of fed compounds and their resulting "processing" by the lacewing males.

**Specific comments:**
**6) In some context, the authors claim is without based on any specific experimental data. The authors have discussed about the "chemical reaction" but have not provided any evidence in support of that. To prove this idea more experimental evidences are required.**

**Response:** This criticism, as well as the next two points, are well taken and, hopefully may be addressed by additional future studies, but cannot be addressed at this point by us. We realize that isotopic labeling experiments are the most precise mechanism to determine biochemical pathways; however, our findings are still pioneering (in our opinion) because: a) our identification of a stereoisomer of iridodial as a pheromone was the first pheromone identified for any lacewing (~1200 species), and the first chemical-baited trapping for any member of the order (~6100 species), b) the discovery that healthy, reproductively competent laboratory-reared males lacked pheromone is an unexpected and important discovery with very practical implications for future research and, despite this Reviewer's claim that our results are not based on any specific experimental data, in fact, c) we showed that feeding the common aphid pheromone components (nepetalactone and nepetalactol, respectively) resulted in finding that laboratory-reared males reduce the lactone to the corresponding dihydrolactone and finding this compound in wild males, and in obtaining the correct stereoisomer of iridodial from lab-reared *C. oculata* males fed nepetalactol, all of which are unequivocally positive results based on experimentation.

**7) To understand the basis of chemical structure determined, the NMR/MALDI-MS data will be required.**

**Response:** As noted herein later in response to Reviewer 3, data was added showing that an individual *C. oculata* male sampled by extraction at a point in time contained about 20 nanograms of iridodial; thus, obtaining enough of this natural product in pure form to

Re: rebuttal & revision of Aldrich et al., "Pharmacophagy in green lacewings (Neuroptera: Chrysopidae: Chrysopa spp.)?"

conduct NMR experiments on the pheromone or intermediates is practically impossible. Nor is MALDI-MS an appropriate or practical analytical approach in this case. In fact, more conventional and precise analytical techniques such as NMR are largely an impractical luxury for insect pheromone analyses due to the limitation of the amount of sample obtainable from insects.

**8) The authors should "confirm" their claim by using a proper "metabolic labelling" experiment where a precursor with radio-labelled/spin-labelled isotope. Such a probe can be added in the food and production of the derivatives in laboratory conditions should be analysed.**

**Response:** Again, unfortunately this approach, especially in our discovery phase of the phenomenon that males evidently must exogenously obtain pheromone precursor(s), was not feasible. The key compounds involved (iridodial, nepetalactone and nepetalactol) each have multiple chiral centers and easily isomerize, making synthetic preparation of labeled isotopes extremely difficult. In fact, the discovery of the aphid pheromone components themselves was facilitated by the revelation that the catnip plant produced the stereochemically correct nepetalactone as found in many aphids. Furthermore, obtaining enough stereochemically correct iridodial to conduct field-trapping experiments has only been possible by utilizing catnip as a source of stereochemically correct nepetalactone. Moreover, at this point we are no longer rearing *C. oculata*, and starting another rearing program to conduct additional experiments, as suggested by the Reviewer, cannot be accomplished in a timely manner.

**9) The possible bio-chemical pathway involved in such process of Pheromone production should be indicated and discussed in details.**

**Response:** This criticism is a valid point, and one that we believe we have addressed by the addition of substantial discussion with up-to-date literature citations:

"Cyclopentanoid natural products based on an iridoid structure are widespread in plants and insects (Hilgraf et al. 2012; Lorenz et al. 1993), and incorporation of [14C]mevalonolactone by the stick insect, Anisomorpha buprestoides (Stoll) (Phasmatodea: Pseudophasmatidae), and the catnip plant (N. cataria) demonstrated that biosynthesis of their respective iridoids, anisomorphal and nepetalactone, proceed via parallel terpene pathways from acyclic precursors, particularly geraniol (Meinwald et al. 1966). Larvae of leaf beetles (Coleoptera: Chrysomelidae) from four different genera showed that biosynthesis of the iridoid defensive compound, chrysomelidial, proceeds from geraniol via an ω-oxidation sequence to 8-hydroxygeraniol, with the eventual cyclization of 8-oxocitral to form the characteristic iridoid cyclopentanoid ring structure (Hilgraf et al. 2012; Lorenz et al. 1993; Veith et al. 1994). Certain rove beetles (Coleoptera: Staphylinidae: Philonthus spp.) also produce defensive secretions containing iridoids (e.g. plagiodial), but unlike enzymes from iridoid-producing leaf beetle larvae, the Philonthus enzyme is able to oxidize and cyclize saturated substrates such as citronellol (Weibel et al. 2001). In plants , including a catnip species (N. racemosa) (Hallahan et al. 1995), the cyclization reactions to iridoids proceed via 10-hydroxygeraniol and 10-oxogeranial rather than 8-hydroxygeraniol/al (Geu-Flores et al.

2012). Furthermore, Hilgraf et al. (2012) stressed that there are still many open questions concerning the biosynthesis of iridoids, particularly "saturated" iridoids such as iridodial."

The feeding studies that we conducted with laboratory-reared *C. oculata* males involved straight-chain monoterpenoid compounds shown in other insects to be suitable precursors for enzymatic cyclization to iridoids. Thus, our interpretation is that the cyclization reaction step is lacking in *Chrysopa* lacewings. Furthermore, the ensuing discussion of pharmacophagy makes it obvious that this type of sequestration is far from being uncommon in insects, apparently even within the Neuroptera (see references to methyl eugenol). What is unique about our study in this regard is that the sequestering organism is a predator rather than a phytophagous insect, and that a prey species (as well as certain plants) are apparently involved in this sequestration system. Finally, there actually is precedent in the literature of iridoid-producing organisms for *de novo* synthesis and sequestration, as we note in our discussion:

"In addition, certain chrysomelid beetle larvae discharge iridoid allomones that may be synthesized *de novo*, which is considered ancestral, or produced via the more evolutionarily advanced mechanism, sequestration from plants (Kunert *et al*. 2008)."

**Reviewer 3 (Anonymous)**

**Comments for the author:**

**This paper reports the goldeneyed lacewing males, *Chrysopa oculata* (Neuroptera: Chrysopidae), which produces (1R,2S,5R,8R)-iridodial as an aggregate from specialized dermal glands on the abdomen. It was also shown that seemingly normal laboratory-reared males of *C. oculata* do not produce iridodial. Feeding studies with *C. oculata* further demonstrate that males of these predatory insects fed one of the common aphid sex pheromone components, (1R,4aS,7S,7aR)-nepetalactol, sequester this compound and convert it to the stereochemically correct lacewing pheromone isomer of iridodial. The experimental designs were mostly carried out correctly. This present manuscript fits well Peer J readership's interests and may become suitable for publications, however, few more additional experiments can covert this manuscript a much better one. The authors should provide quantitative and qualitative data on the pheromone produced by individual animals and also a time point distribution (how much is produced in each day). Identifying the metabolic intermediates is also important.**

**Response:** We are grateful that this Reviewer is largely appreciative of our manuscript! In fact, we did conduct a set of experiments using an internal standard to quantitate how much iridodial can be extracted per wild *C. oculata* male, but we inadvertently neglected to include these data in the originally submitted version of our manuscript. These data, with substantiating raw data files, have been added to the Results section:

"(*Z*)-3-Octen-1-ol was used as an internal standard to quantitate pheromone production per wild *C. oculata* males collected in May 2008; extracts of single males

Re: rebuttal & revision of Aldrich et al., "Pharmacophagy in green lacewings (Neuroptera: Chrysopidae: Chrysopa spp.)?"

contained $20.42 \pm 6.88$ ng iridodial/male (mean $\pm$ SEM; N = 8) (Supplemental Data, Iridodial Quantitation)."